# Injection Moulding into 3D-Printed Plastic Inserts Produced Using the Multi Jet Fusion Method

**DOI:** 10.3390/ma16134747

**Published:** 2023-06-30

**Authors:** Martin Habrman, Zdeněk Chval, Karel Ráž, Ludmila Kučerová, František Hůla

**Affiliations:** 1Faculty of Mechanical Engineering, Regional Technological Institute, University of West Bohemia, Univerzitni 8, 306 14 Plzen, Czech Republic; zchval@fst.zcu.cz (Z.C.); kraz@fst.zcu.cz (K.R.); skal@fst.zcu.cz (L.K.); 23Dees Industries s.r.o., Voctářova 2497/18, 180 00 Praha, Czech Republic; frantisek.hula@3dees.cz

**Keywords:** 3D printing, multi jet fusion (MJF), 3D-printed plastic injection-moulded insert, injection moulding

## Abstract

Most injection-moulded plastics are injection moulded into moulds made from conventional materials such as steel or aluminium. The production costs of the mould are considerable. 3D printing from plastic can be used for injection moulds to save these costs. This article deals with injection moulding into a 3D-printed plastic mould. The injection insert was produced on a HP Multi Jet Fusion 4200 3D printer. The other part of the mould was made of aluminium. A custom injection mould was designed for the research. One insert was made from plastic, and one from aluminium. Both moulds were injected under the same injection conditions. A comparison of injection moulding into the plastic and aluminium inserts is made in this article. The difference when injection moulding into the plastic insert is explained using the different technological conditions. The part injected into the plastic insert was also different from the part injected into the aluminium insert. The difference is explained in this article. This article also looks at the interface between the injection-moulded part and the plastic insert using an electron microscope. The images taken clarify the differences between injection moulding into a plastic insert and an aluminium insert and the differences of the injection-moulded part from the plastic insert.

## 1. Introduction

3D printing is a rapidly developing production method and is a form of rapid prototyping. 3D printing is very often used to replace products manufactured in a conventional way. The replacement of products made of conventional materials with plastic has been in process for a long time. For example, products from steel, aluminium, etc. can be replaced by plastic, etc. In this paper, 3D-printed plastic was used to replace a machined aluminium part in an injection mould [1].

Injection moulding is a conventional way of manufacturing plastic products suitable for mass production. An aluminium mould produced by milling offers a long lifetime but at a higher production price. The use of cheaper plastics and 3D printing to replace more demanding machining will significantly reduce production costs. Cheaper production is beneficial, especially when injection moulding a small number of parts.

Experiments with 3D-printed plastic inserts are currently more focused on production. Many prototype injection moulding companies do not publish their results. Currently, 3D printing methods of photopolymerisation and material jetting are used to produce inserts. 3D printing using selective laser sintering is used in this article [2,3].

The first 3D printing methods were patented 40 years ago. Some patents expired only 10 years ago, which allowed the free use of these 3D printing methods. This led to a great development in 3D printing. The seven basic methods of 3D printing are described by ISO/ASTM 52900. The multi jet fusion (MJF) method was selected for the production of injection-moulded inserts. This is an improved selective laser sintering (SLS) method first patented in 1988 by Carl Deckard. SLS melts powder with a laser. The laser operates in a point area. MJF is based on sintering an entire layer. This method was patented by HP Inc. and is used in their 3D printers. MJF takes place in three steps. In the first step, the fusing agent is applied on the top layer of the powder. In the second step, a detailing agent is applied. In the third step, the powder is sintered in the area where the fusing agent is applied. The fusing agent is a boundary detailing agent, which better defines the boundary of the end of the sintering. Then, the sintered layer is shifted downwards in the unit and more powder is applied. This process is repeated layer by layer. The unsintered powder forms the supports of the parts placed in the space in the unit. Unsintered powder also supports the overhanging 3D-printed parts. The 3D printing is followed by cooling. Compared to other methods, methods using powder are characterised by long cooling times (several hours or even a day). This leads to a high proportion of crystalline phase in their structures [1,4,5].

The use of the MJF method for injection moulding into 3D-printed inserts is relatively new. Compared the MJF to other 3D printing methods used for 3D printing inserts, this method has specific characteristics and distinct 3D-printed parts. The applicability of any 3D printing method in a specific injection moulding scenario is highly dependent on the design of the injection base. For this reason, the mould base used is described in more detail in the following chapter.

## 2. Mould Base

A custom injection mould was designed that was based on our experience of injection moulding experiments in 3D-printed inserts. The plastic material of the mould is not able to withstand contact with the hot nozzle. In addition, the plastic cannot resist high clamping forces. For these reasons and considering the existing designs, the injection mould was divided into a metal frame and a plastic insert, as shown in Figure 1. The insert was 3D-printed from plastic. A space (with the dimensions of the insert) was milled in the metal plates for mounting the insert. The insert was fixed in place by the frame and bolts. This principle is very simple and allows a quick change. The plastic insert with the cavity does not have to fill the entire allocated space. The remaining space can be filled with the supporting inserts (made from metal). The sprue cone was created using a steel insert. The insert in the solid part of the mould had a hole for the sprue bushing [6,7,8].

The uniqueness of the used mould base lies in the space filling (cavity insert and supporting inserts). Minimising 3D printing to only the necessary parts reduces the cost and time required for the production of the insert.

## 3. 3D-Printed Injection-Moulded Insert

The HP Multi Jet Fusion 4200 was used for 3D printing the inserts. The 3D printer can be seen in Figure 2 at the back.

On the left in Figure 2, we can see the processing station where the powder is prepared. The process station is not necessary for 3D printing photopolymerisation and material jetting. The unsintered powder from previous 3D printing jobs was mixed with new powder. Partial powder recycling in 3D printing saves production costs. PA12GB plastic powder is 80% recyclable. The processing station was also used for cooling. Powder and 3D-printed parts were moved from the 3D printer to the processing station using a movable unit. Cooling of the 3D-printed parts in the build unit depended on the cooling mode. The 3D-printed plastic is specified below; the balanced mode was chosen. The total production time for 3D printing and cooling was approximately 2 days [3,4,9].

The plastic used for 3D printing was HP PA12GB. This plastic is filled with 40% glass fibres. Compared to other 3D printing plastics from international company HP Inc., HPPA12 GB (Palo Alto, CA, USA) has the best temperature resistance, dimensional stability, rigidity, etc. These properties are expected of injection moulding inserts. The properties of 3D-printed parts have a significant effect on the application of 3D-printed plastic inserts. The basic properties of the selected plastic are given in Table 1.

Polypropylene (PP) was chosen as the injection-moulded plastic. The advantages of PP include its universal properties, such as its cleaning ability, high permissible shear rate, etc. PP is one of the most widely used plastics for injection moulding. PP 100-GB25 was used from the British multinational chemicals company INEOS Group Ltd (London, UK). The basic properties of this plastic are given in Table 2.

## 4. Comparison of Injection Moulding Inserts for the Experiment

The manufactured injection inserts can be seen in Figure 3. The fixed injection insert for the fixed mould is in the background in Figure 3 and the one for the movable mould is in the foreground. The fixed insert has a hole in the middle for the sprue bushing. Figure 3 shows (a) the aluminium inserts and (b) the HP PA12GB inserts [6].

The aluminium inserts were machined with the standard tolerances for injection moulding. The machining accuracy was selected according to commonly used tolerances for the construction of injection moulds. The roughness of the aluminium surface in the dividing plane was 3.12 µm and the required roughness was 3.2 µm, see Table 3. The roughness was detected with a roughness measuring instrument: HOMMEL-ETAMIC T8000. The measuring range of the TKU300 was 300 µm. The HOMMEL-ETAMIC T8000 was mounted on a Taveline 120 holder with a travel length of 120 mm. The roughness evaluation was performed using the TURBO WAVE V7.45 software.

The roughness of a 3D-printed insert depends on many manufacturing factors. The surface quality of the 3D-printed part depends on its orientation in the build unit. The inserts were located with the cavity faces oriented toward the bottom of the build unit. This is the recommendation of the 3D printer manufacturer in terms of cooling the part. The inserts were placed close to each other so that the shrinkage was more uniform. The roughness of the inserts also depends on the quality of the postprocessing. Unsintered powder remains on the surface of the 3D print, which must be removed by sandblasting. Sandblasting was performed as thoroughly as possible and original HP sand was used for this. An Olympus LEXT confocal microscope was used to measure the roughness of the 3D-printed mould. The measurement was performed in two perpendicular directions. The number of fields of view in each direction was 10. Five shots were taken during the measurement. The analysis was performed by aligning the samples using three points, which aided noise reduction. The λc coefficient was automatically assigned 0.8 mm = 800 μm. The roughness of the evaluation according to the standard ISO 4287 was Ra 10.7 [9].

The difference in surface roughness was obvious. The roughness was one of the parameters that differentiated the plastic inserts from the aluminium ones. Other properties depended on the material used (such as thermal conductivity, see Table 3). The difference was nearly 820 times. This means that plastic was significantly less efficient at conducting heat. This certainly influenced the injection moulding process and the injection-moulded part.

In addition to the technical parameters, economic parameters are also included in Table 3. Preparation time is rounded to days. The 3D printing can begin almost immediately, but it is necessary to set up a production program for machining. If it is possible to start either machining or 3D printing immediately, the time required for production is almost the same. The fundamental difference is in the production price. 3D printing is almost 14 times cheaper [11].

The aluminium insert achieved much better quality than 3D-printed plastic. Due to the lower pressure resistance of the plastic, the size of the 0.45 mm plastic insert was deformed. This modification prevented the creation of the flash.

## 5. Flow Test

To analyse the effect of roughness on the flow of the melt, it was necessary to perform a flow test. Fluidity was analysed in a channel in the path of an Archimedes’ spiral, as seen in Figure 4, left. The channel has a semi-circular profile with a radius of 3 mm. The inserts for aluminium and HP PA12GB were the same and had the same injection conditions. The injection conditions were the same as those shown in Table 4. Milling and 3D printing were the same as for the inserts above. The comparative value was the length of flow. The length into the aluminium insert was 191 mm and the length into the plastic insert was 215 mm. The part injected into the plastic insert is shown in Figure 4. The flash was created by the different distance between the sprue cone and the insert, as can be seen in the centre of the spiral in Figure 4.

Although this distance is within tolerance, it is large enough to cause the flash. The insert with marked lengths was used for this measurement. The length of flow was deduced optically, and the measurement accuracy was 1 mm. The stated result values are the average of 10 measured values. It was found that the length of flow was longer for the insert made from plastic than for the insert made from aluminium. One of the causes of the different lengths could be the air that is on the plastic surface, which prevents the heating of the insert. If the air leaks from the insert, the design of the insert and the mould is correct. If the air is trapped in a cavity, it prevents the melt from flowing [12,13].

It was necessary to analyse the effect of trapped air on the injection properties, e.g., how much the trapped air affects the filling and packing phase. The reason for selecting these two phases is that they are critical to the successful filling of the entire cavity. Both testing methods did not include a packing phase. The first testing mode is the “Pressure limitation”. The principle of testing is to perform the injection moulding up to the pressure limit. When the pressure limit was reached on the injection machine, the filling was stopped. The packing phase was not performed on the injection machine; therefore, the cooling phase followed. The purpose of this test method was to determine how much the surface limited the flow of the plastic material. The result is shown in the graph using a red dashed curve in Figure 5. The second testing mode was labelled “10 s”. As the packing was not set on the injection machine, the filling was set to an action time of 10 s. The longer filling time was to compensate for the packing. The purpose of this test method was to determine how much the trapped air on the surface prevented heat removal from the plastic. The result is show in the graph using the orange curve in Figure 5.

The maximum length of Archimedes’ spiral is longer than the injected plastic was able to fill. The length of flow depends mainly on the injection pressure. The length increases with a higher-pressure value. The pressure applied in the plastic insert cannot be increased too much with regard to its lifetime. The purpose of the following testing was to determine the length of flow for a given pressure with respect to lifetime. If flashes started to appear on the injected parts, the insert was evaluated as bad. The maximum pressure limit for the first method was 500 bar and for the second it was 600 bar. At these pressure levels, flashes started to appear on the injection-moulded parts. The difference was caused by the fact that the plastic has a shorter lifetime if it is in prolonged contact with hot plastic under pressure [14].

The graph in Figure 5 clearly shows the effect of trapped air on the surface. Plastic flow under pressure is only possible if the molten core is not frozen. If a higher injection pressure is applied, a new plastic can be added through a smaller cross section of the molten core. The result can be observed in an increasing length of flow compared to the first method of testing. The length of flow difference between the first and second method is constantly increasing. This difference is constant for metallic materials. Further designs of plastic inserts will have to consider this property.

Furthermore, the graph in Figure 5 determines the maximum pressure at which it is possible to inject into the plastic insert. The safe limit is 400 bars. To achieve greater durability of the insert, it is better to use a lower pressure. The injection pressure was limited to 300 bars.

## 6. Simulation

The following simulation was only performed for the injection moulding into the plastic inserts. The behaviour of the aluminium insert during injection moulding is predictable. The behaviour of plastic inserts, especially with a lower thermal conductivity, is difficult to predict. The set values of the simulation, see Table 4, are the same as the values set on the injection machine.

The simulation was performed using the Mouldex3D software in version 2022. The emphasis in simulating the plastic was on making it as close as possible to real testing. The injection unit was also simulated, which can be seen in Figure 6 [15].

Figure 6 shows the temperature distribution during filling at 0.1 s. For a better illustration of the molten core temperature, Figure 6 shows part of the axonometric view in section. Figure 6 also shows noticeable differences in the mould design. The sprue insert, in Figure 1 in red, was made from steel, and other parts that were in contact with the melt were plastic. Steel removes heat from the melt significantly more than plastic. For this reason, the sprue cone had a lower temperature (blue colour) than the mould cavity (green and red colour), see Figure 6.

The following section focuses on the temperature distribution in the plastic inserts. The experiments with injection moulding into plastic inserts showed that it was more efficient to cool the insert when the mould was opened than during the cooling phase. The cooling time was 40 s. The heat distribution at the start and end of the cooling phase can be seen in Figure 6. There was a uniform scale for both conditions. The ZČU logo shows approximately 90 °C at the start of the cooling phase and approximately 50 °C at the end [7].

Figure 7 shows a detail of the problematic area. The temperature was highest here. Even after 40 s of cooling, this area had a high temperature. However, the temperature of the injected part was low enough that the part could be ejected from the mould.

After the injected part was ejected, the inserts were cooled down after the mould was opened. For faster cooling of the inserts, it is recommended to use compressed air. An opening time after 120 s is shown in Figure 8 to show the temperature distribution.

The next part analyses the problem areas. The packing phase was used to find the problem areas. The cooling phase of the molten core is shown in the time sequence in Figure 9. Gate freezing occurred between 5.4 to 5.7 s. The packing time was correctly set at 6 s. However, the size of the gate was too small. The cavities needed a longer packing time.

## 7. Comparison of Injection-Moulded Parts

Injection moulding into 3D-printed plastic inserts not only changes the injection moulding process, but also affects the moulded parts. Injection moulding was performed under the conditions specified in Table 4. The injection-moulded part (semi-crystalline PP) was transparent. The result of injection moulding into the aluminium insert confirmed this, as can be seen in Figure 10a, where the injection-moulded part has a smooth surface with minimal defects. Figure 10b shows the part injected into the plastic insert. Air traps are only in the channels [11,16].

As expected, injection moulding into an aluminium insert and into a plastic insert was different. The injection-moulded part in the plastic insert was not transparent, because the surface roughness was greater, as seen in Table 3.

## 8. Investigating the Injection-Moulded Part into the Plastic Insert

Microstructure analyses of the injection mouldings and various mould inserts were carried out by light and scanning electron microscopy (SEM) using a Zeiss EVO 25 scanning electron microscope with a LaB6 cathode. The samples for observation were prepared by cutting the inserts with mouldings using a metallographic cutter. The observation and documentation of the surfaces were performed predominantly in back-scattered mode (BSE) [17].

The difference between the injection-moulded plastic (PP) and insert plastic (HP PA12GB) can be seen in Figure 10. The white dots are glass fillings. Both PP and HP PA 12 HB contained glass fillers. While PP had only 25%, HP PA 12 GB had 40%. This percentage difference can be seen in Figure 10 [18].

The difference between the structure of the plastic part and the structure of the mould is hard to distinguish in the lower magnification image. However, the interface of the part and the insert is quite apparent due to the string of holes (dark areas) lining the edges of the injection-moulded part. These areas are the gaps between the injection-moulded part and the insert. This area is created by the shrinkage of the injection-moulded part during cooling [19,20].

The structures can be seen in more detail in Figure 11. The cross section was created by the cross-moulding and the insert Figure 12a is the injection-moulded part and Figure 12b is the insert. Both plastics are semi-crystalline. The 3D-printed plastic cooled for 1 day, allowing enough time to form the semi-crystalline phase. The injection itself only took a few seconds. Thus, the time for crystallisation was shorter and the amount of crystalline phase was less. This can be seen by comparing the two parts of Figure 11. In Figure 12a, there are fine and small particles because the time for creation was short. In Figure 12b, there are coarse and large particles, because the time for creation was longer [12,14].

The difference in structures can be seen at the edge of the injection-moulded part with the insert in Figure 13. The cross section was created by the cross-moulding and the insert.

The injection-moulded part was oriented according to the direction that the cavity was filled. Figure 13 shows a possible cause of the porosity of the 3D-printed plastic insert increasing the roughness.

## 9. Conclusions

The materials used were described in the introduction of the article. Next, the analysis of the manufactured inserts was performed. The analysis performed in preparation before the tests showed significant differences between the aluminium and plastic inserts. The design of the plastic insert depends on the design of the mould base. Mould base designs for injection moulding into 3D-printed plastic moulds are different. This article presents a mould base the design of which was based on our experience with injection moulding 3D-printed plastic inserts. It is certainly a benefit of the approach to fill in the space for inserts.

There are many methods and results that can be performed to investigate injection moulding into 3D-printed inserts. The article focused on methods of investigation using a flow test, simulation, and scanning by electron microscopy. The preparation was performed and described before using these methods.

When examining the interface between the injection-moulded part and the insert, the melt filled the unevenness of the insert surface. A flow test was performed to determine which of the inserts had better flowability. Although the melt caused more irregularities on the surface of the plastic insert, the flow length was longer. One of the possible causes was the trapping of air on the surface. Trapped air works as an insulating layer that prevents cooling of the melt flow and increases the flow length. The roughened surface of the insert causes air to become trapped more easily. Injection-moulded plastic did not flow into the entire cavity with the Archimedes’ spiral. The air did not block the front of the melt. Air is not as easily trapped on the surface of milled aluminium as it is in the 3D-printed plastic insert. Possible future research will focus on analysing how much trapped air on the surface of the 3D-printed plastic insert prevents the ribs from filling, etc.

An insufficient gate size was detected during the packing simulation. However, the same gate size was used for both the plastic and aluminium inserts. The plastic insert and the aluminium insert were, thus, compared under the same conditions. Leaving a large amount of melt in the cavity of the insert caused greater shrinkage of the injected part. This deficiency was worse with the plastic insert than with the steel insert. The reason for this was the inferior heat dissipation of the plastic insert. The simulation of injection moulding into 3D-printed plastic inserts helped to find the disadvantages of this application. When injecting into a plastic insert, it is important that the gate does not freeze early. The simulation also showed a slower heat conduction. Due to the inferior heat dissipation of the plastic insert, the shrinkage problem increased. 

The microscope analysis showed the edge of the injection-moulded part and the plastic insert. The effect of the high roughness of the plastic insert is clearly visible. The effect of slow cooling is also visible.

This article shows the advantages and disadvantages of injection moulding into 3D-printed plastic injection moulding inserts. The published results will certainly help to understand more about this problem. The potential of injection moulding into 3D-printed plastic inserts is great. It can be used in rapid prototyping or small production.

## Figures and Tables

**Figure 1 materials-16-04747-f001:**
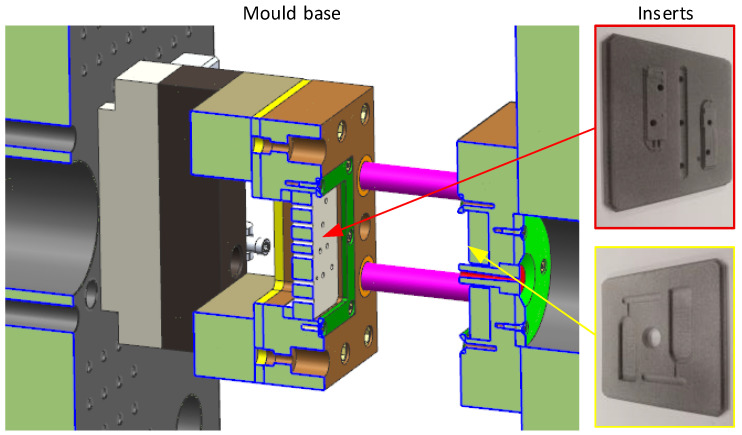
Mould base for testing inserts.

**Figure 2 materials-16-04747-f002:**
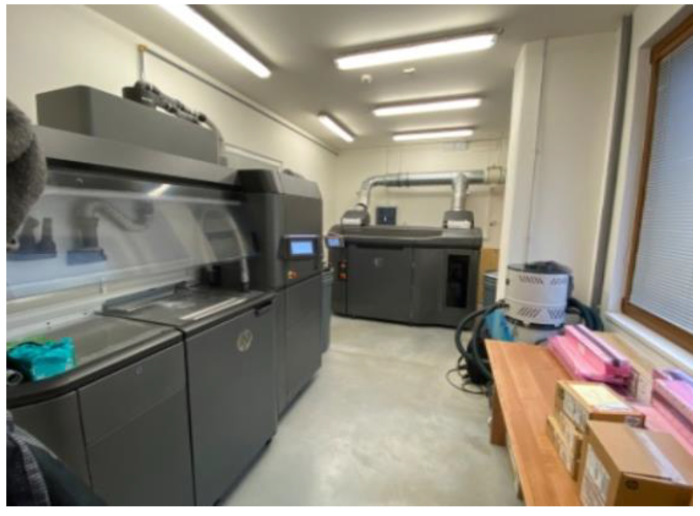
HP Multi Jet Fusion 4200 and processing station.

**Figure 3 materials-16-04747-f003:**
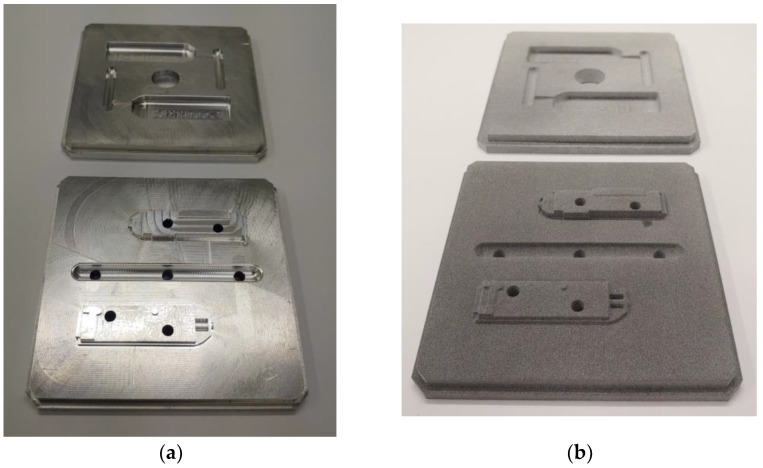
Injection moulding inserts from (**a**) aluminium and (**b**) HP PA12GB.

**Figure 4 materials-16-04747-f004:**
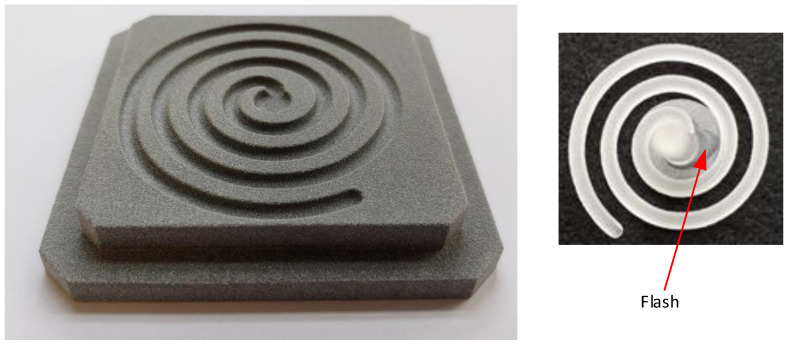
Archimedes’ spiral for testing fluidity and moulded part.

**Figure 5 materials-16-04747-f005:**
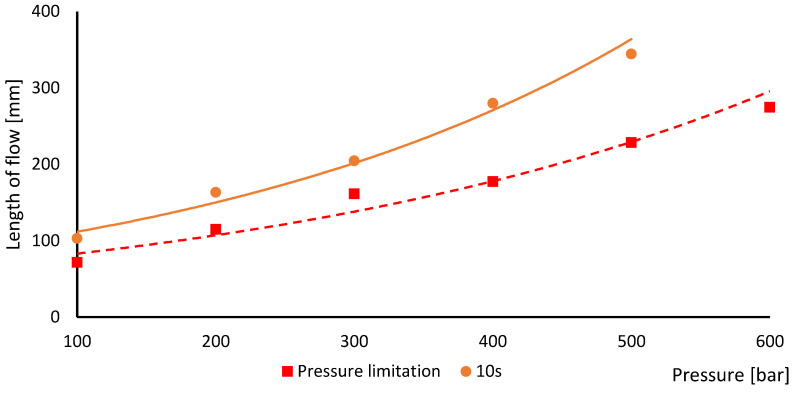
Evaluation of the length of flow in dependence on the injection pressure.

**Figure 6 materials-16-04747-f006:**
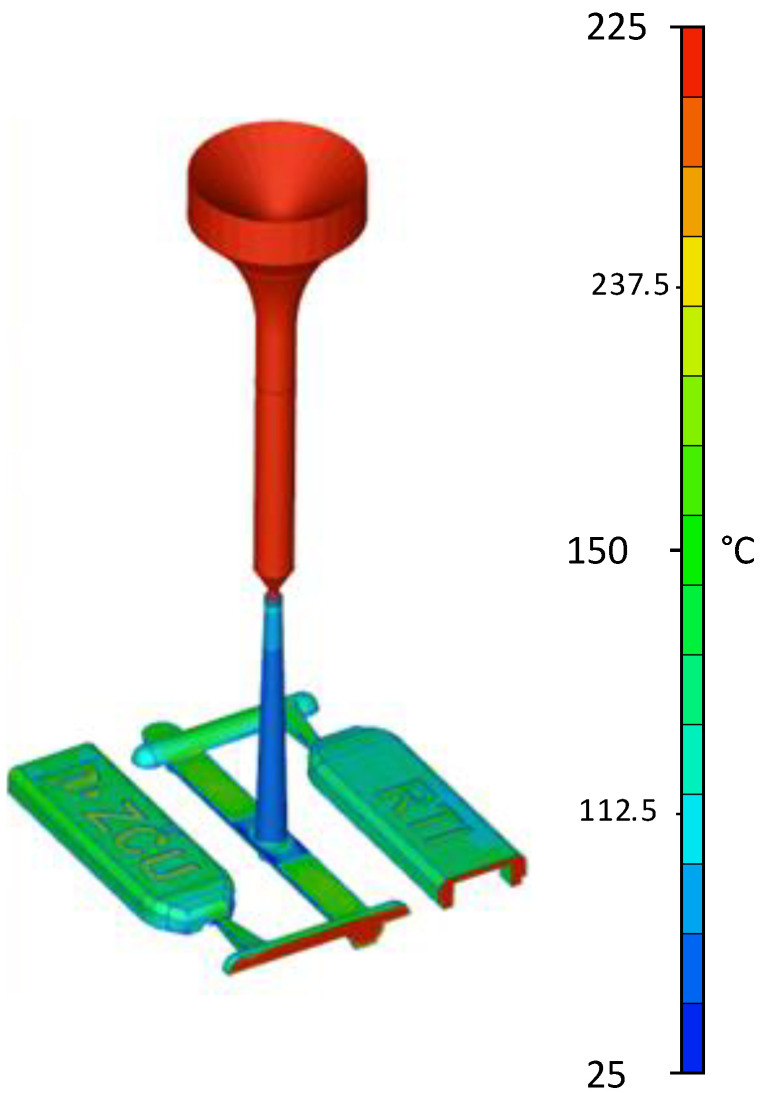
Injection-moulded simulation in axonometric view and in cross section.

**Figure 7 materials-16-04747-f007:**
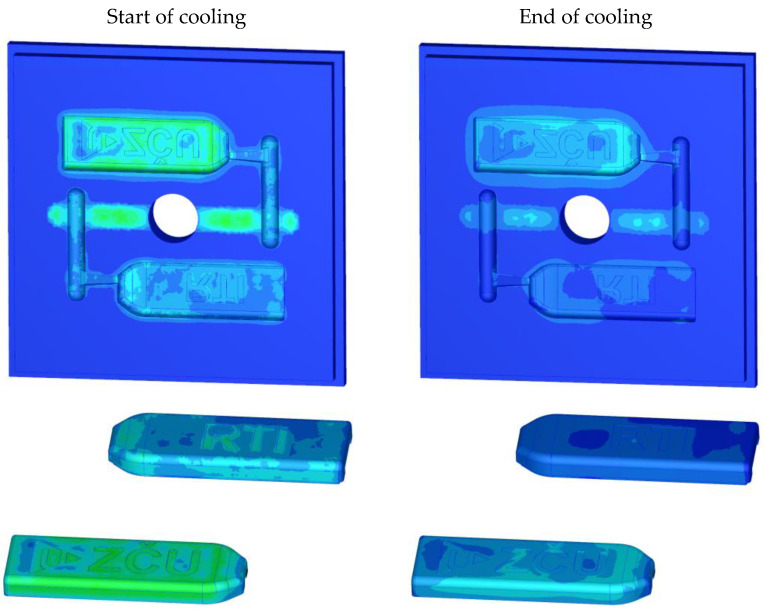
Simulation of inserts during cooling.

**Figure 8 materials-16-04747-f008:**
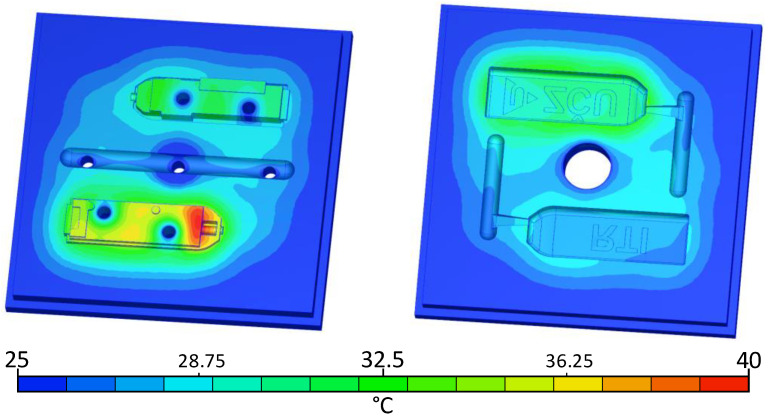
Simulation of inserts during opening time.

**Figure 9 materials-16-04747-f009:**
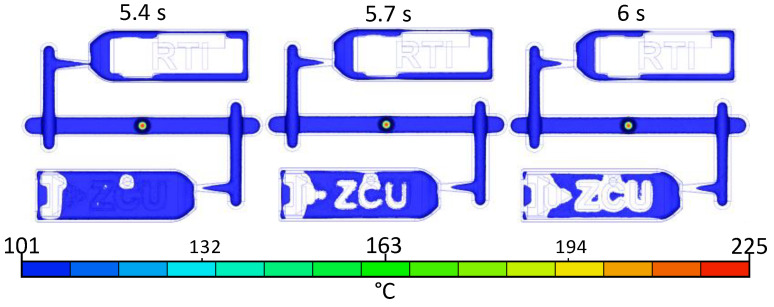
Molten core during packing.

**Figure 10 materials-16-04747-f010:**
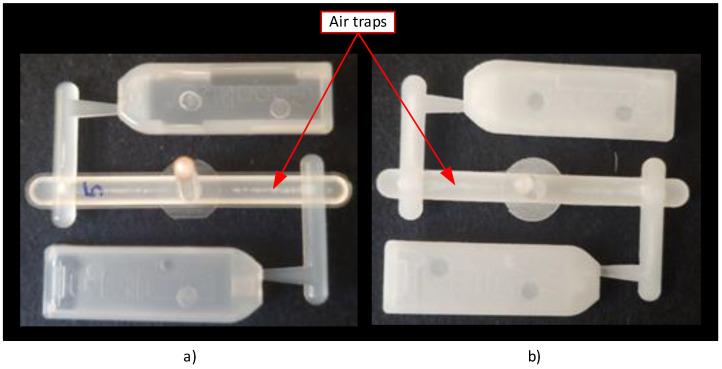
Injection-moulded parts in (**a**) aluminium and (**b**) HP PA 12 GB moulds.

**Figure 11 materials-16-04747-f011:**
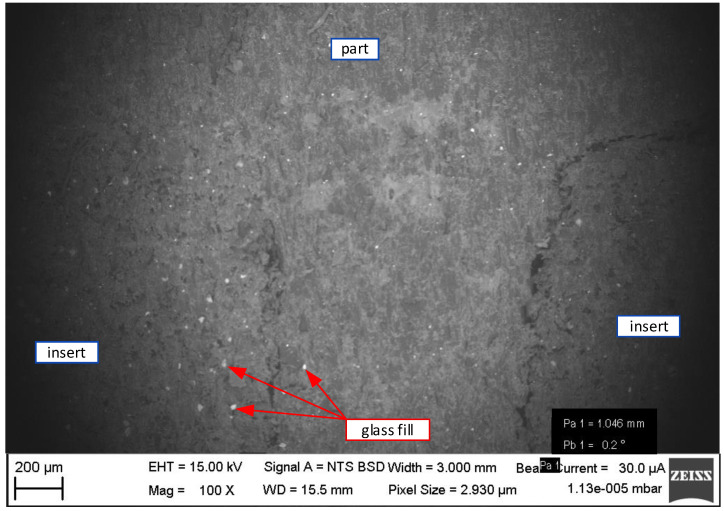
The microscopic photo of the injection-moulded part in the plastic insert.

**Figure 12 materials-16-04747-f012:**
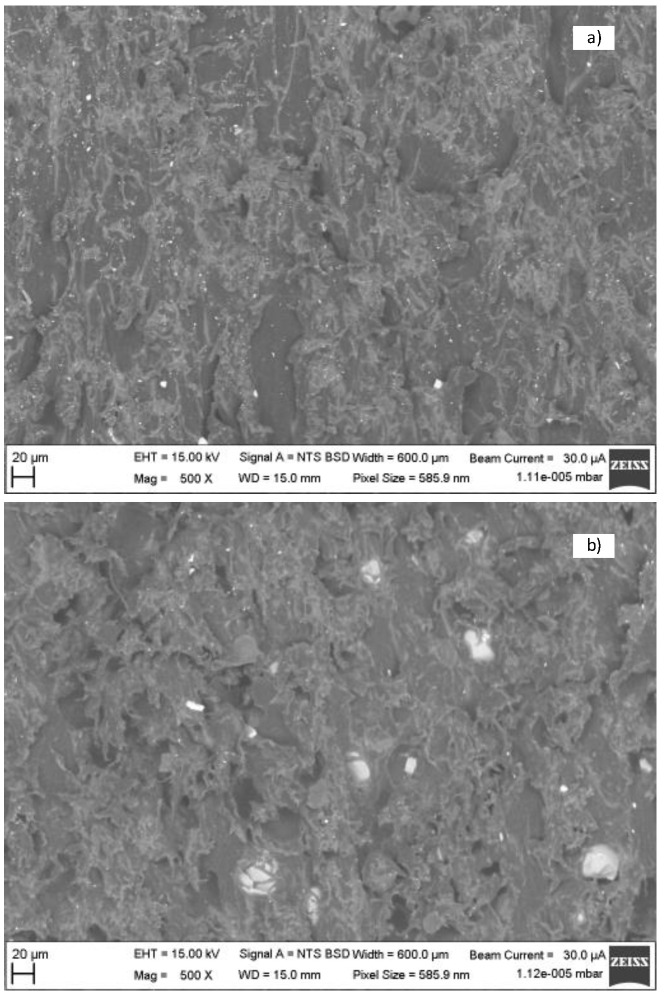
The microscopic photo of the structure of (**a**) injection-moulded part and (**b**) mould insert.

**Figure 13 materials-16-04747-f013:**
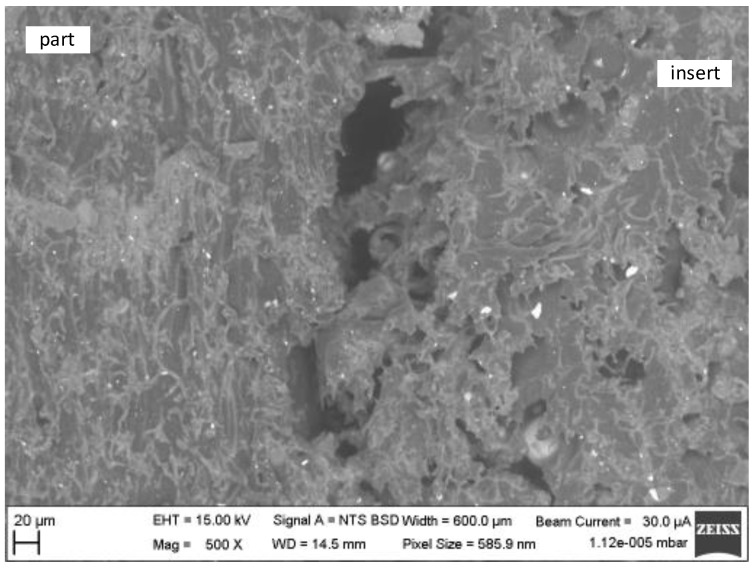
The microscopic photo of the structure of the edge of the injection-moulded part with the insert in detail.

**Table 1 materials-16-04747-t001:** Properties of 3D-printed plastic—HP PA12GB.

Parameter	Test Methods	Value	Unit
General			
Powder melting point (DSC)	ASTM D3418	186	°C
Particle size	ASTM D3451	58	μm
Mechanical			
Tensile strength, max load	ASTM D638	30	MPa
Tensile modulus	ASTM D638	2500	MPa
Thermal			
Heat deflection temperature @0.45 MPa	ASTM D648 Method A	174	°C
Heat deflection temperature @1.82 MPa	ASTM D648 Method A	114	°C

**Table 2 materials-16-04747-t002:** Properties of injection-moulded plastic—PP [10].

Parameter	Test Methods	Value	Unit
Physical			
Melt Flow Rate @230 °C/2.16 kg	ISO 1133	25	g/10 min
Mechanical			
Flexural Modulus @23 °C	ISO 279	1200	MPa
Tensile Strength @Yield	ISO 527-1,-2	32	MPa
Izod impact strength, notched @+23 °C	ISO 180/1A	3.2	kJ/m^2^
Charpy Impact Strength, notched @23 °C	ISO 179/1eA	2.4	kJ/m^2^
Thermal			
HDT @0.45 MPa	ISO 75/B	102	°C
For injection moulding			
Injection temperature—Min		210	°C
Injection temperature—Recommended		225	°C
Injection temperature—Max		240	°C
Mould temperature—Min		31	°C
Mould temperature—Recommended		46	°C
Mould temperature—Max		61	°C
Ejecting temperature		101	°C
Max. shear stress		25,000	Pa
Max. shear rate		100,000	1/s

**Table 3 materials-16-04747-t003:** Comparison of inserts [10].

Parameter	Unit	Aluminium	HP PA 12 GB
Technical properties			
Roughness—Ra	µm	3.12	10.7
Thermal Conductivity	W/(m × K)	205	0.25
Accuracy by		Milling	3D printing
Economic properties			
Preparation time	day	1	0
Production time	day	1	2
Production price	EUR	2 150	105

**Table 4 materials-16-04747-t004:** Injection parameters for simulation and testing.

Parameter	Value	Unit
Volume of the part	6.23	cm^3^
Volume of the injection system	4.73	cm^3^
Filling		
Temperature	225	°C
Pressure	30	MPa
Flow rate	20	cm^3^/s
Volume	10.74	cm^3^
Packing		
Time	6	s
Pressure	21	MPa
Cooling		
Time	40	s
Opening time		
Time	120	s

## Data Availability

The raw/processed data required to reproduce these findings cannot be shared at this time as the data also forms part of an ongoing study.

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
