# Peer review of "Injection Moulding into 3D-Printed Plastic Inserts Produced Using the Multi Jet Fusion Method"

_materials, 2023, doi:10.3390/ma16134747_

Round 1
Reviewer 1 Report
Comments attached

Author Response
Hello,
thank you for the review. Overall, the aim of injection molding into 3D printed plastic inserts is poorly established. Compared to aluminum inserts, they behave completely differently. The process parameters change, the properties of the injection molded part change, etc. I have focused on the specific that are critical for a proper injection moulding into 3D printed ineserts.
I am attaching the corrected version.
Best regards,
M. Habrman
Reviewer 2 Report
The revised version is very good
Author Response
Thank you for your review.
Reviewer 3 Report
The current manuscript investigates the effect of 3D-printed plastic inserts in an injection mould compared to aluminum inserts. The manuscript structure follows a technical report style, not an academic style of a research paper. The current manuscript should be well-revised and organized and the following issues should be considered:
- There should be a critical review of the literature and the problem statement should be well-defined in the introduction section.
- The materials and methods section should be added to specify the applied materials and processes, in addition to illustrating the experimental setup and instruments.
- The results and discussion section should be included to display the obtained results under the title of subsections.
- The current manuscript lacks the discussion and references to justify and validate the obtained results.
- The dimensional accuracy of both the printed and machined inserts should be considered, in addition to comparing the dimensional accuracy obtained from the moulded parts after using each insert material.
- The main contribution and the novelty of the current work are not clear. That should be reflected in the conclusion section.
- The direct applications and future perspective of applying the 3d-printed inserts using the proposed material and printing techniques should be presented.
- It is recommended to use a pullet points style in the conclusion section.
* One more review round is highly recommended after considering the above comments and recommendations.
The language grammar and the selection of the technical words and expressions should be carefully revised.
Author Response

(The authors gave the same response as above.)

Round 2
Reviewer 1 Report
Dear Authors
As no appropriate corrections have been made and adequate explanations have not been provided, I suggest resubmitting the article for review.
Kind regards
Reviewer
Author Response
I tried to present the article in a way that would be interesting for the reader and understand the problems of injection moulding into 3D printed plastic inserts. This is what I have done in the structure of the chapters. In the introduction there is an introduction to the problem. In addition to the 3D printer, the inserts, and their possibilities, it is necessary to also mention the injection mould. There is no injection mould available on the market that is dedicated to injection moulding 3D printed plastic inserts. For this reason, a mould was to be made and a chapter is dedicated to it.
On the flow test I am trying to describe the differences in the behaviour of the plastic insert during injection moulding. This is a technological test, which I have included in the article as well as in practice, before the injection moulding of the part.
1) standard included in references
2) the relevant parts in the figure have been described
3) So that no one has to look for a 3D printer, it is shown. At the same time, the picture shows our workplace
4) You are right it is a 3D printing planes. The values are a bit different than what the manufacturer presents, unfortunately. Since the article doesn't address the rotation of the 3D printed part, the directions don't make any sense. The values are presented in the smallest possible way to represent all directions. Values are given in the smallest possible value to represent all directions in the same way.
5) For this reason, I have chosen the labels a) and b)
6) Yes, roughness is very dependent on sanding. The sand used is from HP. Unfortunately, we don't have more information. To ensure the best surface quality, my colleague sanded the inserts very thoroughly.
7) The literature was citated
8) Yes, it is an axonometric view but at the same time it is in cross-section. The cut was made to better illustrate the molten core of the plastic.
9) The sentence has been completed.
10) Yes, the images that were taken with the microscope were correctly labelled.
11) Yes the edge between the insert and the injection moulded part in the microscope. Corrected.
12) The purpose and focus of the article was stated at the beginning and summarized at the end. I focused on one of the many problems that injection moulding into 3D printed plastic inserts creates. The simulation and experiment I have done has hopefully helped to move this problem further. There are several possibilities where to apply the presented problematics.
Thank you for the nice point-to-point marked with numbers. And I apologize for my first reaction. This is my first-time publishing in this article and I don't have much experience with the process yet.
Best regards,
M. Habrman

Reviewer 3 Report
There is a problem t track the changes in the revised manuscript, there should be illustrated by a different color. Also, there is a reference citation error in the manuscript text.
In addition, the response of the authors did not include the consideration of the review comments and recommendations.
The authors should resubmit the revised manuscript after considering the above comments.
Moderate change is required.
Author Response
I tried to present the article in a way that would be interesting for the reader and understand the problems of injection moulding into 3D printed plastic inserts. This is what I have done in the structure of the chapters. In the introduction there is an introduction to the problem. In addition to the 3D printer, the inserts, and their possibilities, it is necessary to also mention the injection mould. There is no injection mould available on the market that is dedicated to injection moulding 3D printed plastic inserts. For this reason, a mould was to be made and a chapter is dedicated to it.
On the flow test I am trying to describe the differences in the behaviour of the plastic insert during injection moulding. This is a technological test, which I have included in the article as well as in practice, before the injection moulding of the part. Based on the data from the test, I proceeded with the injection moulding of the part. I acknowledge that, given the range of issues, this is a larger introduction.
Please where more discussion and references are needed?
Successfully performing injection molding into plastic 3D printed inserts is very complicated nowadays. The aim of the article was to approach the successful application of this problem. Unfortunately, the problem is so large that there is not enough in one article.
I tried to include the revisionts in the article. I apologize for my first reaction. This is my first-time publishing in this journal and I don't have much experience with the process yet.
Best regards,
M. Habrman

Round 3
Reviewer 1 Report
Dear Authors,
After the corrections made and the explanations presented, I have no further comments.
Kind regards
Reviewer
Author Response
Thank you for your evaluation.
One from reviewers found problems that led me to make changes in the article.
I resubmit the article again just to be sure.

Reviewer 3 Report
The revised manuscript presents some interesting results that should be revised and re-organized after considering the following issues and comments:
. The manuscript structure seems as a technical report, please review the Journal instructions to the authors to organize the manuscript in an academic paper structure.
- The introduction still lacks a critical review from the literature, in addition to clearly define the problem statement by the end of this section.
- The materials and methods section should be added to present the characteristics of the applied materials, processes, instruments, and the experimental procedures.
- The results and discussion section should include the current chapters as subsections to display and validate the obtained results. The discussion should be supported by more references to justify the results.
- What about the durability of the 3d printed plastic parts and the quality of products compared to aluminum parts?
- The direct applications and future perspective of the proposed method should be presented.
- It is highly recommended to use a pullet points style to focus on the main results, contributions, and novelty of the current study..
A thorough revision of grammar and word selection should be conducted.
Author Response
1) recommended structure - materials and method
3D printed parts are highly dependent on the 3D printing method. For this reason, the 3D printed is presented, including the 3D printer and Processing station. Next, I need to describe the mould base, because its design is influenced by 3D printing. In addition to the material, it is necessary to compare the manufactured parts in one chapter.
There are many methods. In this article I focused on Flow test, Simulating and investigating the injection moulded.
// The correction was made
2) The literature has been repaired
3) The structure of the article and the division into chapters is due to the more complex problems
4) Added a note about the modification, which indicates that the plastic has a lower resistance to pressure than the aluminium insert.
5) Future perspective is great, added a note to the conslusiuon.
6) The pullet points style not been used but it is explained on Flow test, Simulation and Microscope analysis the purpose of this article.
